**Data Availability Statement:** All relevant data are within the paper and its Supporting information files.

**Funding:** The University of Gondar provides minimal financial support. The funder has no role

# Prevalence and risk factor for mistreatment in childbirth: In health facilities of Gondar city, Ethiopia

**Dagmawit Shemelis[1], Abebaw Addis Gelagay[2], Moges Muluneh Boke [2]***

**1** University of Gondar Specialized Hospital, Gondar, Ethiopia, **2** Department of Reproductive Health, Institute of Public Health, College of Medicine and Health Sciences, University of Gondar, Gondar, Ethiopia

* mogelove75@gmail.com

## Abstract

### Background

Providing compassionate and respectful maternity care to mothers is a vital intervention to improve health outcomes of women and newborns. However, there is less data of compassionate and respectful maternity care in Gondar city. Therefore, this study aimed to assess the magnitude of mistreatment and associated factors among mothers who gave birth at the public health facilities in Gondar city, northwest Ethiopia.

### Methods

Institutional based cross-sectional study was conducted from March to April 2019 in Gondar city public nine health facilities. A total of 584 randomly selected women in the postpartum period were recruited in this study. A binary logistic regression analysis was done to see whether there was an association between mistreatment and independent variables. Finally, the logistic regression analysis was done by stratifying type of parity and mode of delivery.

### Results

Overall, 73.2% (95% CI: 69.7–76.7%) of the women were mistreated during their childbirth care. Non-consented care was the most commonly experienced form of mistreatment (63.6%, 95% CI: 59.6–67.6%). Having less than four antenatal care follow-up visits (AOR = 3.58, 95% CI: 2.04–6.29), giving birth in the hospital (AOR = 2.83, 95% CI: 1.52–5.27), and facing complications during delivery (AOR = 2.06, 95% CI: 1.52–3.98) were significantly associated with mistreatment among postpartum mothers.

### Conclusions

This study showed a lower proportion of mistreatment than other studies in Ethiopia. Having less than four ANC follow up, place of current delivery, and facing complication during delivery were identified as the determinants of mistreatment. Therefore, this calls for strengthening actions, like providing maternity education during antenatal care and appropriate management of complications to improve the quality of maternity care at health facilities,

in the conception, design, analysis, interpretation, and decision on publication.

**Competing interests:** The authors have declared that no competing interests exist.

and enhancing hospital working health workers capacity on compassionate and respectful maternity care.

## Introduction

Maternal mistreatment at childbirth can be unintentional neglect due to staffing constraints, or intentional physical abuse, related to the behavior of individuals like verbal abuse, and sexual abuse, or health facility related constrain like shortage of beds and failure to meet professional standards of care [1]. Many women experience mistreatment during childbirth globally, in multi-country study four in ten women experienced physical or verbal abuse, or stigma and discrimination mistreatment at health facilities-based birth [2]. The prevalence of disrespect and abuse at birth vary region to region. In European it ranges from 13.5%-30.2% [3], in Latin America 8%-66.8% [4], and in Africa 0.6%-98.9% [1, 5–8]. In Ethiopia the situation is similar to Africa, it ranges from 78%-98.9% [5, 9].

Maternal mortality is the death of a woman at period of prenatal or within 42 days of termination of pregnancy, regardless of the site and duration of the pregnancy, from any cause related to or provoked by the pregnancy or its treatment, but not include accidental or incidental causes [10]. Globally, more than half of a million women die annually as a result of pregnancy and childbirth-related complications. In sub-Saharan Africa the maternal mortality ratio (MMR) is 510 per 100, 000 live births, it accounts 61% of the global maternal deaths [11]. Ethiopia is one of the nations with the highest maternal mortality ratio. The maternal mortality ratio in Ethiopia was 412 per 100,000 live births [12]. Moreover, the country needs to triple the former 2.3% annual maternal mortality reduction rate to attain the newly set Sustainable Development Goal (SDG) three target one; stated as reducing global maternal mortality to less than 70 per 100,000 live births by 2030 [13].

Global initiatives develop strategies to minimize maternal mortality by improving the proportion of professional births and ensuring that all women have access to high-quality reproductive health services (3). Mistreatment of women during childbirth at health facilities violates their human rights and autonomy and also highly relate to preventable maternal and newborn mortality and morbidity [14].

The World Health Organization (WHO) defines compassionate and respectful maternal care (CRC) as the right every woman has to attain the highest standard of health; includes the right to dignity, compassionate and respectful health care to all childbearing women around the world throughout pregnancy, childbirth, and the postnatal period [15]. Mistreatment is any form of inhumane treatment or uncaring behavior toward a woman in facility-based during childbirth [16]. A laboring mother is subjected to a diverse form of mistreatment such as physical abuse, lack of consent for care, non-confidential care, undignified care, abandonment, discrimination, and detention in facilities for failure to pay user fees [8]. Mistreatment of maternal care is one of the causes of maternal mortality and morbidity through hindering maternal health care service utilization. Many women have been mistreated during childbirth in healthcare facilities around the world, as a result, they refuse to use maternal healthcare services or return to those facilities [16]. Therefore, alleviating mistreatment is important to improve human rights and decrease the risk of consequences such as postpartum depression and to advance health service utilization [17–19].

Compassionate and respectful service during childbirth is a key component to improving maternity care quality, healthcare efficiency, and maternal mortality and morbidity reduction

(8). Over the past half-decade, the Ethiopian government has delivered CRC training to ensure health care quality (9). However, three out of four mothers give birth at home without a qualified health professional, and four hundred twenty mothers out of one hundred thousand live births die as a result of pregnancy-related complications and care [12, 20]. In Amhara region the coverage of antenatal care (67%) and institutional deliveries (27%) were low compared to Addis Ababa (96.8%, 92.4%), Tigray (90%, 56.9%), Harare region (75.9%, 52%), respectively. In Gondar city, there is also a significant difference between antenatal care coverage and skilled birth coverage, this disparity suggests that poor revisiting experiences for maternity care [12].

In pervious few of studies being young age, no or low educational status, low economic status, higher parity, mode of delivery, low antenatal care visits women were more likely exposed to mistreatment during childbirth [1, 5–8].

In Ethiopia, health centers are centers for maternity care, despite the fact that the majority of studies conducted on childbirth care disrespect and abuse has been mainly focused on the hospital setting only. There is also a limitation of disrespect and abuse of childbirth care evidence in Gondar. Therefore, this study aimed to estimate the prevalence of mistreatment and associated factors among postpartum women who gave birth at the public health facilities in Gondar city, Northwest Ethiopia.

## Methods

### Study design and period

An institutional-based cross-sectional study was carried out at the health facilities of Gondar city from March to April 2019.

### Study area and population

The study was conducted in the public health facilities of Gondar city, Northwest Ethiopia. Gondar city is in the North Gondar zone of Amhara National Regional State, Northwest Ethiopia which is located 737 km far from the capital city, Addis Ababa. Gondar has a total population of 338,646, according to the Gondar city woreda administration bureau report, in which 49.8% are females. Gondar city has eight public health centers, 14 health posts, 32 private clinics, and one referral hospital. Of those, eight health centers and the referral hospital that have been providing childbirth care were included in this study. The coverage of institutional deliveries and antenatal care was 62%, and 86%, respectively in 2011 EFY (2019 G.C). The study was conducted in eight health centers (Azezo health center, Gondar health center, Merak health center, Woleka health center, Teda health center, Genbot 20 health center, Biajing health center, and Gebreale health center) and one referral hospital (Gondar University specialized hospital).

All postpartum mothers who gave birth at public health facilities of Gondar city during the data collection period were the study population.

### Sample size determination and sampling procedure

The sample size was estimated using a single proportion formula with the following assumptions: 95% level of confidence, 4% margin of error, 10% non-response rate, and 67% of mistreatment during childbirth taken from a study conducted in Bahir Dar town, Ethiopia [21], which yields a total of 584.

In this study, all childbirth care providing health facilities; eight health centers, and one referral hospital were considered. Gondar University specialized hospital, Azezo health center,

**Table 1. Operational definition of mistreatment for postpartum mothers who give birth at public health facilities in Gondar City, northwest Ethiopia, 2019 (n = 574).**

| Variables | Definition | Reference |
|---|---|---|
| Mistreatment | If a mother experienced at least one of the seven categories of mistreatment (physical abuse, lack of consent for care, non-confidential care, undignified care, abandonment, discrimination, and detention in facilities), she was considered as "mistreated " | [16] |
| Physical abuse | Is physical force or abrasive behavior with the woman, including slapping or hitting and touches measured using six criteria. A woman who answers yes to at least one criterion was considered as physical abuse at the time of labor and delivery. | [16] |
| No confidential care | Lack of confidentiality and lack of privacy during maternal care measured using two criteria. A woman who answers yes to at least one of the criteria then she was considered as exposed to non-confidential care. | [6] |
| Non consented care | Absence of informed consent, or patient communication, forced procedure, measured using seven criteria, A Woman who answers yes to at least one criterion was considered as being mistreated at the time of labor and delivery. | [6] |
| Non-dignified care | Is measured using two criteria: lack of dignity, respect, and intentionally humiliating, scolding, or shouting at women. A woman who answers yes to at least one of the criteria was considered as being mistreated at the time of labor and delivery. | [6] |
| Discrimination | Is the luck of equitable care which is measured using three criteria. A woman who answers yes to at least one criterion was considered as being mistreated at the time of labor and delivery. | [6] |
| Abandonment or denial of care | Is a lack of right to timely healthcare and the highest attainable level of health, measured using two criteria. A woman who answers yes to at least one of the criteria then she was considered as being mistreated at the time of labor and delivery. | [6] |
| Detention in facilities | Is detaining of mothers in a health facility is measured using two criteria: deprivation of liberty, autonomy, self-determination, and coercion. A woman who answers yes to at least one of the criteria was considered as being mistreated at the time of labor and delivery. | [6] |

Gondar health center, Merak health center, Woleka health center, Teda health center, Genbot 20 health center, Biajing health center, and Gebreale health center had averagely 928, 66, 38, 40, 25, 50, 11,7,8 skilled births per month respectively [22]. First, the estimated sample size was proportionally allocated to the eight health centers and one referral hospital based on their monthly deliveries number. Then, a systematic sampling technique with two k intervals was used to select the study participants at each health facility.

## Operational definition

Mistreatment was assessed by seven categories of disrespect and abuse tools. That is, a total of 24 verification criteria of mistreatment were used to evaluation. Each category of disrespect and abuse care were defined as follow, see (Table 1). Other variables were defined as follow, age was measured by year, it categorized into five groups such as 15–19 years, 20–24 years, 25–29 years, 30–34 years and 35 years and above. Education was assessed based in level of education: no formal education, elementary (grade 1–8), secondary school (grade 9–12), and college and above. Residence was classified as urban or rural. Marital status was categorized in to single, married and divorced. Religion considered as Orthodox, Catholic, Protestant and Muslim. Parity was classified as primiparous and multiparous. Mode of delivery was categorized as cesarean section and vaginal delivery. ANC visit was categorized as less than four and four and above. Place of delivery for index birth was categorized as hospital and health center. Facing complication during labour categorized as yes and no. Time to stay in the health facility was analyzed as less than 12 hours, 12–24 hours and more than 24 hours.

## Data collection tool and procedures

An interviewer-administered structured questionnaire was developed after reviewing different studies and program tool kit (11). The tool was initially developed in English and translated into the local language (*Amharic*), and finally back to English to ensure consistency. The questionnaire has sections; socio-demographic characteristics, obstetric characteristics, and mistreatment assessments. Mistreatment was assessed by seven categories of disrespect and abuse

tools. The mistreatment tool was developed from the Maternal and Child Health Integrated Program (MCHIP) respectful maternity care tool kit [23].

Data were collected through face-to-face interviews and chart review was done to collect obstetric related variables. Six trained diploma nurses and two BSc graduated nurses from the nearby Tseda health science college were recruited as data collectors and supervisors, respectively. A pretest was conducted on 5% of the sample in Loza Mulu health center (out of the study setting). The supervisor checked the data collection process daily and overall supervision was done by the principal investigator.

To ensure data quality, a two days training was given to data collectors and supervisors on objectives of the study, data collection instruments, techniques, and procedures. The data collectors were supervised daily, and the consistency and completeness of the data were checked by the principal investigator every night.

## Statistical analysis

Data were cleaned and checked for completeness and consistency before they were entered into Epi-info version 7.2 statistical software and exported to SPSS windows version 20.0 software for analysis.

Descriptive statistics were expressed using summary statistics such as mean, range, frequency, and percentage. Confidence intervals were drawn for each mistreatment proportion using SPSS software. Both bivariable and multivariable logistic regression analyses were computed to identify the determinants. All explanatory variables with a p-value of less than 0.25 in the bivariable logistic regression analysis were included in the final multivariable logistic regression model after checking multicollinearity assumptions. Moreover, the model fitness was checked using Hosmer and Lemeshow's test (P = 0.135). In the final model, a p-value of less than 0.05 and Adjusted Odds Ratios (AOR) with a 95% confidence interval (CI) were used to identify statistically associated factors.

## Ethics approval and consent to participants

Ethical clearance (Ref. No. IPH/180/06/2011) was obtained from the Ethical review board of the Institute of public health, college of medicine and health Sciences, University of Gondar's Ethical Review board. Then a formal letter of cooperation was written from the Gondar city administration health office to health centres. The purpose of the study was clearly explained to the study participants. After the purpose and objective of the study have been informed, written consent was obtained from each study participant. Participants were also being informed that participation was voluntary and they can stop or leave the participation at any time if they are not comfortable. To keep confidentiality of any information provided by the study subjects, the data collection procedure was anonymous and keeping their privacy during the interview by interviewing them alone. According to the Ethiopian national research ethics review guideline, consent from parents or guardians for adolescents aged less than 18 years is waivered for some cases. Among these emancipated/mature minors, one is, those who married or established their own family [24]. Hence, consent was not obtained from parents or guardians for these pregnant adolescents.

## Results

### Socio-demographic characteristics of study participants

A total of 574 women responded to the interviewer-administered questionnaire, with a response rate of 98.3%. The mean age of the participants was 28 (SD = 5.76) years. Nearly one-

**Table 2. Socio-demographic characteristics of mothers who gave birth at public health facilities in Gondar city, Northwest, Ethiopia, 2019 (n = 574).**

| Types of variables | Category | Frequency | Percent |
|---|---|---|---|
| Age of mother | 15–19 | 31 | 5.4 |
| | 20–24 | 133 | 23.2 |
| | 25–29 | 181 | 31.5 |
| | 30–34 | 133 | 23.2 |
| | > = 35 | 96 | 16.7 |
| Mean age (SD) | | 28 (SD = 5.76) | |
| Marital status | Single | 25 | 4.4 |
| | Married | 543 | 94.6 |
| | Divorced / Widowed | 6 | 1.0 |
| Mother's religion | Orthodox | 453 | 78.9 |
| | Catholic | 6 | 1.0 |
| | Protestant | 8 | 1.4 |
| | Muslim | 107 | 18.6 |
| Mother's level of education | Not read and write | 99 | 17.2 |
| | Read and write | 46 | 8.0 |
| | Primary (1–6) | 37 | 6.4 |
| | Junior (7–8) | 52 | 9.1 |
| | Secondary (9–12) | 199 | 34.7 |
| | Collage and above | 141 | 24.6 |
| Mother's occupation | Housewife | 362 | 63.1 |
| | Private employee | 36 | 6.3 |
| | Government employee | 103 | 17.9 |
| | Merchant | 31 | 5.4 |
| | Student | 30 | 5.2 |
| | Other* | 12 | 2.1 |
| Place of residence | Urban | 374 | 65.2 |
| | Rural | 200 | 34.8 |
| Family monthly income in Ethiopian birr | 1st quartile | 223 | 38.9 |
| | 2nd quartile | 143 | 24.9 |
| | 3rd quartile | 208 | 36.2 |

Note
* = daily labor.

third of the participants were in the age group of 25–29 years. The majority (94.6%) of the study participants were married; 78.9% were Orthodox Christian followers; 63.1%were housewives, and 17.2% never attended formal education (Table 2).

## Obstetric history of postpartum mothers

From the total number of participants, 29.6% were primigravida, more than half (56.3%) had a previous history of institutional childbirth. The majority (81.4%) of the study participants had at least one ANC visit, (9.4%) had a history of abortion, and (15.7%) had a history of stillbirth. Almost all participants (98%) had at least one ANC visit during the recent pregnancy. More than one-fourth (78%) of mothers gave birth via the spontaneous vagina. Nearly half, of the mothers, stayed less than 12 hours in the health facility. Nearly seventy-nine percent of the participants have received labor and delivery services in hospitals. Six percent of mothers have faced hypertension during labor and delivery (Table 3).

**Table 3. Obstetric characteristics of mothers who gave birth at public health facilities in Gondar city, Northwest, Ethiopia, 2019 (n = 574).**

| Types of variable | Category | Frequency | Percent % |
|---|---|---|---|
| Parity | Primiparous | 199 | 34.7 |
| | Multiparous | 336 | 58.5 |
| | Grand multiparous | 39 | 6.8 |
| Gravidity | One | 170 | 29.6 |
| | Two-four | 276 | 48.1 |
| | Five and above | 128 | 22.3 |
| History of abortion | No | 521 | 90.8 |
| | Yes | 53 | 9.2 |
| History of stillbirth | No | 483 | 84.1 |
| | Yes | 91 | 15.9 |
| ANC visit for the previous birth, n = 404 | No | 75 | 18.6 |
| | Yes | 329 | 81.4 |
| History of previous institutional delivery, n = 402 | No | 86 | 21.4 |
| | Yes | 316 | 78.6 |
| Number of ANC follow up for current child | I had no followed up | 11 | 1.9 |
| | One follow up | 10 | 1.7 |
| | Two follow up | 49 | 8.5 |
| | Three follow up | 133 | 23.2 |
| | Four and above | 371 | 64.6 |
| Mode of delivery | Vaginal delivery | 448 | 78 |
| | Cesarean delivery | 126 | 22 |
| Length of stay in the health facility | Less than 12 hours | 278 | 48.4 |
| | 12–24 hours | 155 | 27.0 |
| | More than 24 hours | 141 | 24.6 |
| Mode of discharge | With professional's recommendation | 566 | 98.6 |
| | Self (AMA) | 8 | 1.3 |
| Place of receiving ANC | Health post | 32 | 5.6 |
| | Health center | 295 | 51.4 |
| | Public hospital | 265 | 46.2 |
| | Private clinic | 68 | 11.8 |
| HCP conducting ANC follow up | Doctor | 283 | 49.3 |
| | Nurse | 10 | 1.7 |
| | Midwife | 438 | 76.3 |
| | HEWs | 32 | 5.6 |
| Complication during delivery | No | 423 | 73.6 |
| | Yes (for Mother) | 73 | 12.7 |
| | Yes (for baby) | 67 | 11.7 |
| | Yes (both mother and baby) | 11 | 1.9 |
| Types of facing complication for mothers (73) | Hypertension | 34 | 46.5 |
| | Postpartum hemorrhage | 14 | 19.1 |
| | Prolonged labor | 11 | 15.0 |
| | Antepartum hemorrhage | 9 | 12.3 |
| | Other complications* | 5 | 6.8 |

(*Continued*)

**Table 3.** (Continued)

| Types of variable | Category | Frequency | Percent % |
|---|---|---|---|
| Types of complication for new born baby (67) | Fetal distress | 31 | 46.2 |
| | Low birth Weight | 16 | 23.8 |
| | Preterm birth | 6 | 8.9 |
| | Abnormal presentation | 8 | 11.9 |
| | Other complication** | 6 | 8.9 |

Note

*, other complication for mother is post-term, hypothermia, oligohydramnios, wound infection after CS delivery;

**, Other complications for baby are jaundice, birth injury and fiver; ANC, antenatal care; HCP, health care provider; AMA, against medical advice.

## Proportion of mistreatment during facility-based childbirth

From a total of 574 participants, nearly three-fourths (73.2%, 95%CI: 69.7 to 76.7%) participants were reported at least one form of mistreatment during facility-based childbirth (Fig 1).

## Type of mistreatment during facility-based childbirth

We counted mothers who experienced at least one condition among the possibilities based on verification requirements for categories of mistreatment. Accordingly, 171(29.8%, 95% CI: 25.4 to 32.8%) women were not protected from physical harm or ill-treatment during childbirth. The other most commonly experienced form of mistreatment was the prohibition of information, informed consent, and choice/preferences 365(63.6%, 95% CI: 59.6 to 67.6%). The commonly violated criterion under this domain was the provider did not respond mother's question with politeness and truthfulness 520(90.6%) and the providers did not explain what is being done and what to expect throughout labor and birth 513(89.4%). The second commonly reported type of mistreatment was no -confidentiality care 171(29.8%, 95% CI: 26.0 to 33.6%). Commonly violated criterion under this domain was the health care providers did not use drapes or covering appropriate to protect the mother's privacy 415(72.3%). In addition to these, 89(14.6%, 95% CI: 11.7 to 17.4%), women reported undignified care. The majority of 67(11.7%) of respondents reported under this domain were health providers shouted at or scolded during labor and delivery. On the other hand, from the total participants, 76(11.7%, 95% CI: 9.2 to 14.6%) women experienced neglect care during labor and delivery. Under this domain commonly reported criteria were health providers ignored or abandoned when postpartum women called for help 58(10.1%) (Fig 1), (S1 Table).

## Factors associated with mistreatment during facility childbirth

In multivariable logistic regression analysis, the number of ANC visits for the recent pregnancy, type of health facility attended for delivery, and having complications during delivery were factors statistically associated with mistreatment.

Accordingly, mothers who had less than four ANC follow up were 3.58 times (AOR = 3.58, 95% CI: 2.04, 6.29) more likely experienced mistreatment than those who had four and more ANC visits. Mothers who gave birth in the hospital were 2.83 times more likely to experienced mistreatment than those who gave birth in health centers (AOR = 2.83, 95% CI: 1.52, 5.27). Besides, mothers who faced complications during delivery were 2.06 times more likely to experienced mistreatment than those who did not face complications (AOR = 2.06, 95% CI: 1.52, 3.98) (Table 4).

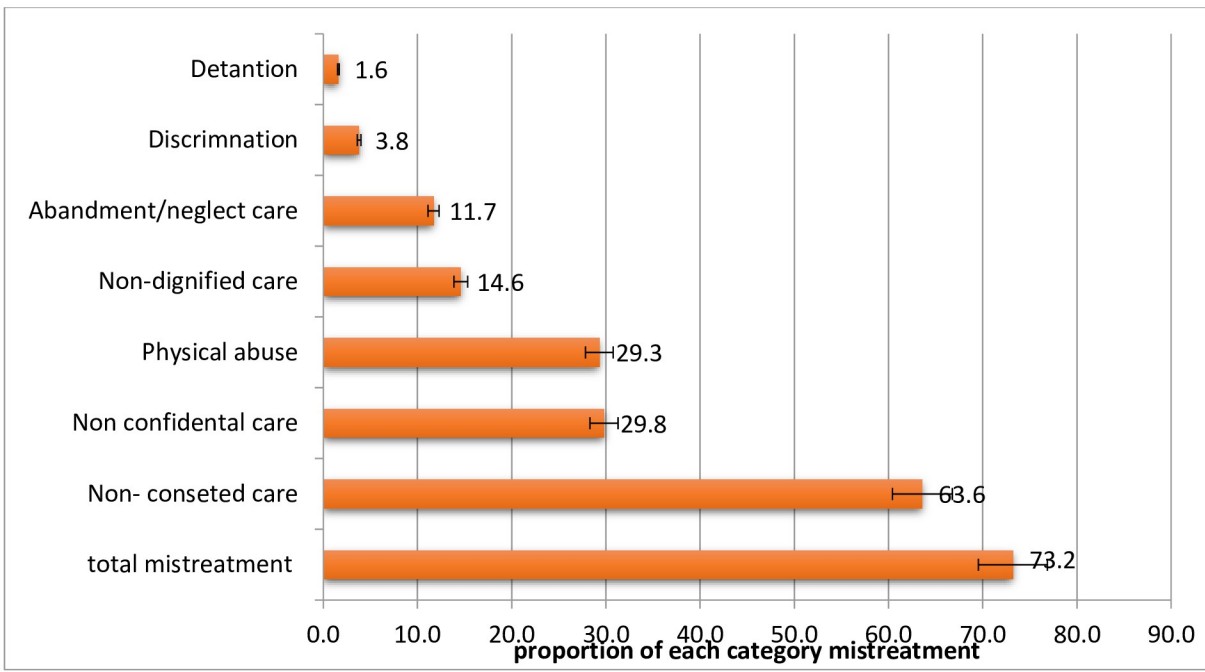

**Fig 1. Proportion of mistreatment by category during facility base childbirth in Gondar city, 2019.**

The analysis was stratified based on parity and mode of delivery. Less than minimum recommended Antenatal care visits or less than four visits (AOR = 3.84, 95%CI: 1.94–7.59), facing complication during delivery (AOR = 2.50, 95%CI: 1.09–5.69), and hospital place of delivery (AOR = 3.07, 95%CI: 1.40–6.70) were associated with an increased likelihood of mistreatment during childbirth among women who were multiparous, in contrast, facing complication during delivery (AOR = 1.66, 95%CI: 0.54–5.07), and hospital place of delivery (AOR = 1.99, 95% CI: 0.71–5.59) were not associated with mistreatment among women with primiparous (Table 5).

Regarding stratified analysis based on mode of delivery, less than minimum recommended Antenatal care visits (AOR = 3.59, 95% CI: 1.89–6.83) was associated with an increased odds of mistreatment during childbirth among women delivered with vaginal, but it did not associate among cesarean section delivery women (AOR = 3.55, 95% CI: 0.98–12.52) (Table 6).

## Discussion

This study investigated the proportion of mistreatment among postpartum mothers who gave birth at health facilities. Besides, the study aimed to identify factors that were associated with mistreatment among postpartum mothers who gave birth at health facilities.

Overall, 73.2% of women who gave birth at health facilities were mistreated. This finding is lower than these of studies conducted in Addis Ababa (78%) [25] and Arba Minch (98.9%) [5]. The discrepancy might be attributed due to the study period, study population, and measurement differences; recently the issue of compassionate and respectful maternity care has given more attention. Besides the study participants, variation in the proportion of the previous history of institutional delivery, in the current study 21% of the study participants had no previous history of institutional delivery, while in studies conducted Addis Ababa (31%), and Arba Minch (37%). The absence of a previous history of institutional delivery was reported as a risk factor for childbirth care mistreatment in the current study. Moreover, the other possible

**Table 4. Bivariate and multivariate logistic regression analyses of mistreatment during facility childbirth (N = 574).**

| Types of variables | Experienced mistreatment | | COR (95% CI) | AOR (95% CI) |
|---|---|---|---|---|
| | No | Yes | | |
| **Mother age** | | | | |
| 15–19 years | 9 | 22 | 1 | 1 |
| 20–24 years | 36 | 97 | 1.10 (0.46–2.62) | 1.22 (0.48–3.13) |
| 25–29 years | 59 | 122 | 0.85 (0.37–1.95) | 1.39 (0.53–3.69) |
| 30–34 years | 34 | 99 | 1.19 (0.50–2.84) | 1.76(0.62–5.02) |
| > = 35 years | 16 | 80 | 2.04 (0.79–5.25) | 2.17(0.68–6.98) |
| **Religion** | | | | |
| Orthodox | 114 | 339 | 1.32 (0.84–2.10) | 1.32 (0.79–2.20) |
| Catholic | 3 | 3 | 0.45 (0.08–2.33) | 0.82 (0.13–5.01) |
| Protestant | 4 | 4 | 0.45 (0.10–1.89) | 0.73 (0.10–3.38) |
| Muslim | 33 | 74 | 1 | 1 |
| **Educational status** | | | | |
| No formal education | 24 | 121 | 2.61(1.39–4.28) | 1.32(0.79–2.20) |
| Elementary 1–8 | 14 | 75 | 2.68(1.33–5.07) | 0.82(0.13–5.01) |
| Secondary 9–12 | 70 | 129 | 2.84(0.56–1.41) | 0.73(0.10–3.38) |
| College and above | 46 | 95 | 1 | 1 |
| **Residence** | | | | |
| Urban | 119 | 255 | 1 | 1 |
| Rural | 35 | 165 | 2.16(1.44–3.36) | 0.99(0.56–1.76) |
| **ANC visit** | | | | |
| Less than four visits | 23 | 180 | 4.27(2.63–6.93) | 3.58(2.04–6.29) * |
| Four and above visit | 131 | 240 | 1 | 1 |
| **Facing complication during labor** | | | | |
| No | 134 | 311 | 1 | 1 |
| Yes | 20 | 109 | 2.35(1.39–3.94) | 2.06(1.52–3.98) * |
| **History of abortion** | | | | |
| No | 98 | 253 | 1 | 1 |
| Yes | 7 | 46 | 2.58(1.17–5.99) | 1.36(.055–3.36) |
| **Place of delivery for index** | | | | |
| Hospital | 138 | 315 | 2.87(1.64–5.05) | 2.83(1.52–5.27) * |
| Health center | 16 | 105 | 1 | 1 |
| **Time to stay in the health facility** | | | | |
| Less than 12 hours | 89 | 189 | 1 | 1 |
| 12–24 hours | 35 | 120 | 1.61(1.03–2.54) | 1.15(0.69–1.92) |
| More than 24 hours | 36 | 111 | 1.74(1.08–2.80) | 1.02(0.36–2.90) |
| **Parity** | | | | |
| Primiparous | 53 | 146 | 1.01(0.68–1.49) | 1.56(0.89–2.73) |
| Multiparous | 101 | 274 | 1 | 1 |
| **Mode of delivery** | | | | |
| Vaginal | 126 | 322 | 0.73(0.46–1.17) | 1.03(0.39–2.75) |
| Cesarean section | 28 | 98 | 1 | 1 |

Note

* Reminded the significance of the variable (P-value <0.05).

**Table 5. Bivariate and multivariate logistic regression analysis stratification by parity.**

| Types of variables | Primiparous (N = 199) | | Multiparous (N = 375) | |
|---|---|---|---|---|
| Maltreatment (Yes) | COR (95% CI) | AOR (95% CI) | COR (95% CI) | AOR (95% CI) |
| **Educational status** | | | | |
| No formal education | 1.53(0.31–7.68) | 0.18(0.01–2.60) | 3.05(1.77–5.26) | 1.34(0.65–2.78) |
| Elementary 1–8 | 1.98(0.41–9.50) | 1.31(0.24–7.11) | 3.24(1.07–9.80) | 1.03(0.63–6.49) |
| Secondary 9–12 | 5.05(1.14–22.3) | 3.10(0.68–14.62) | 2.06(0.79–5.36) | 1.47(0.53–4.08) |
| College and above | 1 | 1 | 1 | 1 |
| **Residence** | | | | |
| Urban | 1 | 1 | 1 | 1 |
| Rural | 5.06(1.48–17.2) | 6.76(0.85–53.90) | 1.92(1.18–3.11) | 0.77(0.41–1.48) |
| **ANC visit** | | | | |
| Less than four visits | 4.14(1.54–11.1) | 3.88(1.38–10.9)* | 4.54(2.59–7.97) | 3.84(1.94–7.59)* |
| Four and above visit | 1 | 1 | 1 | 1 |
| **Facing complication during labor** | | | | |
| No | 1 | 1 | 1 | 1 |
| Yes | 1.94(0.76–4.98) | 1.66(0.54–5.07) | 2.56(1.38–4.77) | 2.50(1.09–5.69) * |
| **Place of delivery for index birth** | | | | |
| Hospital | 2.47(0.97–6.26) | 1.99(0.71–5.59) | 3.12(1.54–6.33) | 3.07(1.40–6.70)* |
| Health center | 1 | 1 | 1 | 1 |
| **Time to stay in the health facility** | | | | |
| Less than 12 hours | 1 | 1 | 1 | 1 |
| 12–24 hours | 1.59(0.79–3.42) | 1.32(0.59–2.94) | 1.64(0.89–2.84) | 0.96(0.50–1.85) |
| More than 24 hours | 1.75(0.73–4.04) | 1.26(0.45–3.52) | 1.78(0.98–3.11) | 0.82(0.37–1.82) |

**Table 6. Bivariate and multivariate logistic regression analyses stratification by mode of delivery.**

| Mistreatment | Vaginal delivery (N = 448) | | Cesarean section (N = 126) | |
|---|---|---|---|---|
| | COR (95% CI) | AOR (95% CI) | COR (95% CI) | AOR (95% CI) |
| **Mother age** | | | | |
| 15–19 | 1 | 1 | 1 | 1 |
| 20–24 | 0.61(0.21–1.79) | 0.77(0.25–2.35) | 9.00(1.20–67.42) | 7.29(0.81–65.38) |
| 25–29 | 0.45(0.16–1.28) | 0.60(0.20–1.79) | 6.60(1.05–41.51) | 6.17(0.79–47.64) |
| 30–34 | 0.65(0.22–1.91) | 0.71(0.23–2.22) | 9.20(1.30–64.8) | 6.10(0.71–52.83) |
| > = 35 | 1.32(0.41–4.26) | 1.08(0.30–3.84) | 8.80(1.24–62.19) | 5.20(0.47–57.82) |
| **Educational status** | | | | |
| No education | 2.79(1.57–4.97) | 1.08(0.50–2.32) | 1.89(0.71–5.06) | 0.67(0.13–3.31) |
| Elementary 1–8 | 3.46(1.17–10.22) | 2.71(0.88–8.26) | 1.13(0.21–6.17) | 0.50(0.07–3.33) |
| Secondary 9–12 | 2.95(1.19–7.29) | 2.49(0.98–6.34) | 2.27(0.46–9.19) | 1.34(0.23–7.98) |
| College and above | 1 | 1 | 1 | 1 |
| **Residence** | | | | |
| Urban | 1 | 1 | 1 | 1 |
| Rural | 2.13(1.29–3.52) | 1.34(0.73–2.45) | 2.06(0.87–4.86) | 0.98(0.25–3.81) |
| **ANC visit** | | | | |
| Less than four visits | 4.63(2.58–8.31) | 3.59(1.89–6.83)* | 3.33(1.34–8.30) | 3.55(0.98–12.52) |
| Four and above visit | 1 | | 1 | 1 |
| **Facing complication during labor** | | | | |
| No | 1 | 1 | 1 | 1 |
| Yes | 3.33(1.34–8.30) | 2.10(0.92–4.77) | 2.51(1.07–5.91) | 2.40(0.89–6.49) |

explanation for the discrepancy might be the differences in the measurements of mistreatment. In the study conducted in Addis Ababa mistreatment was measured using six categories while in the current study seven categories (physical abuse, lack of consent for care, non-confidential care, undignified care, abandonment, discrimination, and detention in facilities) were used. Whereas the disparity with that of a study conducted in Arba Minch's may be attributed to the inclusion of outdated techniques as physical violence, such as using fundal pressure to expel babies and suturing episiotomies or perineal tears without the use of local anesthesia.

On the other hand, the proportion of mistreatment was higher in this study compared to the studies conducted in Africa [14–20%] [6, 7, 26]. This variation might be due to the difference in data collection techniques, for example, in studies conducted in Malawi and Tanzania, observation was used to collect data that might decrease the finding because of the observation bias/hawthorn effects during the procedures.

In our study, non-consented care is the most commonly experienced component of mistreatment which was found to be 63.6%. The finding is higher compared to that of a study done in Addis Ababa and showed that 48% of respondents were experienced non-consented care [25]. Similarly, this finding was higher than the same study done in Kenya 4.3% [7]. This inconsistency might be due to the difference in health policy and implementation program.

Our finding showed that 29.8% of women experienced non-confidentiality care. This finding is higher than the same study done in Nigeria (26%) and Kenya (8.5%) [7, 27]. In our setting, maintaining privacy in the studied health facilities is a critical challenge. This may be due to the presence of a large number of mothers with a limited number of admission beds and an inadequate waiting room, which may have interfered with women's privacy, especially in the hospital.

Our study highlighted that 29.3% of women experienced physical abuse during labor and delivery. This result is lower than these of studies conducted in Nigeria (35.7%) and Ethiopia (32.9%) and lower than Bahir Dar 57.6% [21, 25, 27]. This shows that physical abuse has been persistently high in different countries. This may be because, in low-resource settings, laboring women perceive physical violence in health facilities as a normal occurrence during the second stage of labour.

The study finding showed that 14.6% of women had got non-dignified care. This finding is lower than these of studies conducted in Nigeria (29.6%) and Kenya (18%) [7, 27]. This discrepancy might be due to the difference in study period and area.

The current study reported that 11.7% of the women faced abandonment/neglect care during labor and delivery. This finding is lower than that of a study conducted in Addis Ababa, Ethiopia, and showed that 39.3% faced neglected care and higher than that of a study conducted in Tanzania 8% [25, 26].

Similarly, the other category of mistreatment reported in this study was discrimination during the provision of service which accounts 3.8%. This finding is lower than that of a study conducted in Nigeria and showed that discrimination was 20% [27]. The inconsistency may be due to socio-cultural differences and the presence of committed health care providers in our setup.

In this study, the odds of mistreatment were higher among participants who had less than four ANC visits during current pregnancy compared to those who had four and above ANC follow-up. This finding is similar to these of studies conducted in Tanzania and Bahir Dar [21, 26]. These may be those who are new to the environment (less than four ANC follow up) have difficulty coping with the environment can easily perceive them as mistreated.

The study showed that mothers who faced complications during delivery were more likely to mistreat than those who had no complication. This study was consistent with that of a study conducted in India [28]. A high patient load at the facilities, especially at a higher-level facility

where the majority of the complicated cases were referred, which might have contributed in increasing the abusive behavior of health workers.

In this study, mothers who had to deliver in the hospital were more likely to mistreat than those who gave birth at health centers. This finding is similar to these of studies conducted in Arba Minch and Addis Ababa, Ethiopia [5, 25]. This may be due to the hospital staff's workload and stress; hospital staffs are responsible for a large number of clients as well as an emergency.

Moreover, this study findings reported that distinct pattern of association depending on mode of delivery and type of parity. The effect of facing complication, ANC visits, and place of delivery exposure increased the risk of mistreatment during childbirth was higher among multiparous women. The effect of ANC visits on mistreatment occurrence was also significantly higher in vaginal delivery mode.

In this study, information was gathered immediately after childbirth which reduces the potential of recall bias and it estimated the overall prevalence of disrespect and abuse during childbirth care from multiple health centers and hospitals that highlights the strengths of the study. The limitation of this study is mistreatment measurement tool was developed from the Maternal and Child Health Integrated Program respectful maternity care tool kit which was not validated in Ethiopia and not assess health facility related constrains. Also, this study is conducted in a health facility setting hence a potential for selection bias. In this study the data collection is immediately after delivery, women might not be report all violence freely due to suffer consequences (more mistreatment) from health providers and gratitude bias. Another limitation of this study is fundal pressure and episiotomy procedure were not excluded which might be inflate the prevalence mistreatment.

## Conclusion

The result of this study revealed that three in every four mothers were subjected to mistreatment during health facility childbirth care. Non-consented care and non-confidential care were the most prevalent form of mistreatment. ANC visit, type of health facility visited for childbirth care, and facing complication during labour and delivery, were significantly associated with health facilities childbirth care mistreatment. Therefore, this calls for strengthening actions, like providing maternity education during antenatal care and appropriate management of complications to improve the quality of maternity care at health facilities. Also, health care providers that work in childbirth care need to be trained on the importance of informed consent and give compassionate and respectful care. Moreover, Health facilities need to promote positive birth experiences through the provision of respectful, dignified, supportive, and consented care.

## Supporting information

**S1 Table.**
(PDF)

**S1 Questionnaire.**
(DOCX)

**S1 Checklist. STROBE statement—Checklist of items that should be included in reports of *cross-sectional studies*.**
(DOC)

## Acknowledgments

I would like to forward my deepest appreciation and thanks to Gondar zonal health department and the respective health facility administration office for facilitating the data collection process and all data collectors, supervisors and respondents for their collaboration for the success of this study. Finally, would like to thank the University of Gondar, Institute of Public Health & Medical Sciences for giving this chance.

## Author Contributions

**Conceptualization:** Dagmawit Shemelis.

**Data curation:** Dagmawit Shemelis.

**Formal analysis:** Dagmawit Shemelis, Abebaw Addis Gelagay, Moges Muluneh Boke.

**Methodology:** Dagmawit Shemelis, Abebaw Addis Gelagay, Moges Muluneh Boke.

**Supervision:** Abebaw Addis Gelagay, Moges Muluneh Boke.

**Writing – original draft:** Dagmawit Shemelis, Abebaw Addis Gelagay, Moges Muluneh Boke.

**Writing – review & editing:** Dagmawit Shemelis, Abebaw Addis Gelagay, Moges Muluneh Boke.

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
