## [Decision Letter · Decision Letter 0]

16 Mar 2021

PONE-D-21-04196

High attribution of non-consented care for high prevalence of mistreatment among postpartum mothers who give birth at public health facilities in Gondar City, northwest Ethiopia

PLOS ONE

Dear Dr. Boke,

Thank you for submitting your manuscript to PLOS ONE. After careful consideration, we feel that it has merit but does not fully meet PLOS ONE’s publication criteria as it currently stands. Therefore, we invite you to submit a revised version of the manuscript that addresses the points raised during the review process.

We look forward to receiving your revised manuscript.

Kind regards,

Orvalho Augusto, MD, MPH

Academic Editor

PLOS ONE

Additional Editor Comments:

Improving maternal child care access requires the quality of care and respectful care. The authors document the levels of women mistreatment during delivery in the Gondar City. Their data and account will contribute to the relatively scarce literature for the vast sub-Saharan Africa region.

However, there are some important issues that need to be addressed before this report is acepted.

1. Please correct the many English typos. Here are few examples:

line 63 - Experian's

line 67 - Review the sentence around the "Ethiopian government"

lines 69 to 71 it is unclear. May be it is better to write the number using numeric symbols

lines 72 to 74 please mention the name of the study area. By this moment the reader does not know what study are is.

line 75 put a period after "gap". Remove the "and indicate". Put "It aims to estimate".

Line 89 after “Gondar city” there should be a verb

Line 129 after “questionnaire” it should be “was developed”

lines 245 and 246 put the confidence interval in the brackets

line 321 what the word “ever” means here. This is repeated in the whole conclusion as well as in the abstract. Please review.

It would be highly beneficial to find some English review support.

2. Careful with the interpretation of the reference. In the first paragraph of the introduction you have reference 1 (The 2014 UNICEF report on maternal mortal between 1990 and 2013). Please find that report to correct your numbers. There is no, in that report, an African MMR. There is a sub-Saharan Africa estimate. And it is 510 per 100,000 live-births; also the fraction of deaths is not 64%, is 61%. [Check that report on table 2 and on page 21].

3. In the description of the study area please do mention the private clinics as well. Also, do add basic description of coverage of institutional deliveries and antenatal care. This information is useful to understand the group included.

4. On sampling procedures please use words to explain what is that k-interval for the systematic sampling. See further comments of the reviewer below.

5. The operational definition section needs a table with the list of things to make these definitions. It would be beneficial if the English instrument were added in the supplements.

6. Data processing and analysis

This section needs to be expanded.

- We do not analyze a logistic regression. We use a logistic regression to analyze something. Please correct.

- There is use of confidence intervals for a proportion in the results. How such thing interval was estimated. This is something to put in the methods.

- Line 219 is for the “data processing and analysis” section not in the “results”. Please correct.

7. Results see the reviewer comment.

- See the comment about the line 219

- When report the mean as in the line 163 we usually write the SD as, for example, “(SD = 5.76)”. Notice the equals sign.

- Remove the “The” on “The proportion of mistreatment during facility-based childbirth”

- Please report the confidence intervals for all mistreatment types in text

8. table 1

- Add the mean and SD of the age

- Make all percentages 1 decimal place

- Add more categories for the “family estimated monthly income” variable. And please provide quartiles.

- On “HCP conducting ANC follow up” variable please correct the “Privet clinic”. That should be “Private clinic”.

- Please check the alignment for “complication during delivery”.

9. Table 3

- for the ORs please make all 2 decimal places

- Make sure that references have 1 (for example it is missing on Educational status)

- Why age and “religion” are not included? Regardless of the p-value. This is an key variable.

10. Figure 1

- This pie is taking a lot of space to inform exactly the same in text. And it does not provide any confidence interval as in the text you did. Drop it. I suggest to add this information on figure 2.

11. Figure 2

- Add the proportion of mistreatment you have on pie. Add this to the bar plot.

- Add to all of these bar error bars to capture the confidence interval

- Fix the decimal places to 1

- Remove the percentage symbol in the plot

- Remove the so called main title this can be placed in the x-axis and in caption of the plot

12. Discussion

- Lines 253 to 257. The reason for the differences did you check that these studies define the same way as you do here? Are the populations comparable besides the health facility? Be very careful. We do not want to label medical schools as the center of mistreatment.

- What the authors thing of the tool they used to assess the mistreatment ? Did you check for validity to use in this setting? Please discuss this. And most likely this must be considered for the limitation subsection.

- The strengths and limitations need more elaboration. This is a health facility based study therefore a potential for selection bias may doom the generalization of this study. There is always possibility of confounding this must be discussed.

13. Please fill the STROBE statement and add to the supplements (https://www.strobe-statement.org/index.php?id=strobe-home)

14. Add a specific and clear section of ethics

15. Abstract

- OR please make sure they have 2 decimal places

- For all mistreatment proportion make sure they have confidence intervals

3. In your Methods section, please provide additional information about the participant recruitment method in particular a description of how participants were recruited.

Please include copies of the survey questions or questionnaires used in the study, in both the original language and English, as Supporting Information, or include a citation if they have been published previously.

4. You indicated that you had ethical approval for your study. In your Methods section, please ensure you have also stated whether you obtained consent from parents or guardians of the minors included in the study or whether the research ethics committee or IRB specifically waived the need for their consent.

5. We suggest you thoroughly copyedit your manuscript for language usage, spelling, and grammar. If you do not know anyone who can help you do this, you may wish to consider employing a professional scientific editing service.  

7. Thank you for submitting the above manuscript to PLOS ONE. During our internal evaluation of the manuscript, we found significant text overlap between your submission and the following previously published works:

- https://www.biorxiv.org/content/10.1101/430199v1 (Introduction, paragraph 1, sentences 3-5) (Introduction, paragraph 4, sentences 1-2) (Discussion, paragraph 13, sentence 3)

- https://www.tandfonline.com/doi/full/10.1080/09688080.2018.1502020 (Introduction, paragraph 2, sentence 1)

- https://www.researchsquare.com/article/rs-107805/v1 (Introduction, paragraph 2, sentence 2)

- https://preview-reproductive-health-journal.biomedcentral.com/articles/10.1186/s12978-015-0024-9 (Types of mistreatment during facility-based childbirth, sentences 1-2) (Discussion, paragraph 5, sentence 4)

- https://journals.plos.org/plosone/article?id=10.1371%2Fjournal.pone.0205545 (Discussion, paragraph 2, sentence 4) (Discussion, paragraph 12, sentence 4)

- https://obgyn.onlinelibrary.wiley.com/doi/abs/10.1016/j.ijgo.2014.08.015 (Discussion, paragraph 6, sentence 4)

- https://bmcpregnancychildbirth.biomedcentral.com/articles/10.1186/s12884-018-1970-3 (Discussion, paragraph 11, sentence 4)

Please revise the manuscript to rephrase the duplicated text, cite your sources, and provide details as to how the current manuscript advances on previous work. Please note that further consideration is dependent on the submission of a manuscript that addresses these concerns about the overlap in text with published work.

Reviewers' comments:

Reviewer's Responses to Questions

**Comments to the Author**

1. Is the manuscript technically sound, and do the data support the conclusions?

Reviewer #1: Yes

2. Has the statistical analysis been performed appropriately and rigorously? 

Reviewer #1: Yes

3. Have the authors made all data underlying the findings in their manuscript fully available?

Reviewer #1: No

4. Is the manuscript presented in an intelligible fashion and written in standard English?

Reviewer #1: No

5. Review Comments to the Author

Reviewer #1: Thank you for the opportunity to review this paper.

It is an important topic on an issue that is prevalent in LMICs including Africa

General comments:

1. Grammar and spelling: The paper is written in poor English, has multiple grammatical and spelling errors throughout the entire document. I have highlighted most of them in the manuscript. Recommendation: the authors need to rewrite the paper in standard English to enable the readers to understand.

2. Justification(line 72-74): the two statements contradict each other. Which is the correct position?

3. Study site: it is not clear whether the study was conducted in one or multiple study sites. What is the name of the facility? How was it chosen from among the other 14 facilities in the city? What are the numbers of deliveries handled by this facility per annum? What made this site the most ideal for your study?

4. Sample size: the explanation after sample size calculation is not clear. recommendation: edit for clarity

5. Data collection procedures: Review the whole of this section for consistency in grammar eg the use of "is" instead of "was". Clarify the role of the degree nurses vis a vis the principal investigator. What research experience did the six research assistants have? Were the research assistants employees of the same hospitals or were they independently employed for the study only? how did you address the issue of bias if these nurses worked in the same facilities where they conducted the interviews? Lots of redundancies and repetitions eg questionnaire was pretested (written more than twice)

6. Results: clarify if n=574 or 584(all tables). Table 2: In some of the cells eg previous ANC attendance, the "n" is not 574 as some of the women were carrying a first pregnancy

7. Discussion: some of the explanations are not coherent eg why pregnant women who delivered in a hospital before were likely to be mistreated. Review the explanations

8. Strengths and limitations: too brief and not clear

9. Conclusions: Edit for clarity

10. Ethics and consent: did the study pose any risks to the study participants? what care was offered to the abused women? were they linked to any care? Did you obtain parental consent for those below 18 years of age. Edit the grammar for clarity

6. PLOS authors have the option to publish the peer review history of their article (what does this mean?). If published, this will include your full peer review and any attached files.

Reviewer #1: **Yes: **George Gwako

---

## [Author Response · Author response to Decision Letter 0]

14 May 2021

To: Editor-in-chief of PLOS ONE Journal 

Subject: Submission of a revised manuscript for publication

Dear Editor,

Thank you for allowing us to submit a revised draft of our manuscript entitled “High attribution of non-consented care for high prevalence of mistreatment among postpartum mothers who give birth at public health facilities in Gondar City, northwest Ethiopia "[Manuscript ID number: PONE-D-21-04196]. We appreciate the time and effort dedicated by you and the reviewers to provide your valuable feedback on our manuscript. We are grateful to the reviewers for their insightful comments which improve our manuscript.

We accepted and tried to incorporate all of the comments provided. Thus, the comments are attached here below with their point-by-point responses given in blue font color. Besides, the detailed changes made are highlighted in the “revised manuscript with track changes” to easily identify the changes/improvements, and also the clean copy of the revised manuscript, questionnaire and STROBE form are prepared. On edition of manuscript all authors, and Tsagye Haile from Belgium, assistant professor of health policy was involved. 

Response to Editor

1. Please correct the many English typos

 Response 1: We authors tried to improve the language including the typos

2. Careful with the interpretation of the reference. In the first paragraph of the introduction, you have reference 1 (The 2014 UNICEF report on maternal mortal between 1990 and 2013). Please find that report to correct your numbers. There is no, in that report, an African MMR. There is a sub-Saharan Africa estimate. And it is 510 per 100,000 live-births; also the fraction of deaths is not 64%, is 61%. [Check that report on table 2 and on page 21].

Response 2: Thank you dear editor, we have checked the reference and evidence documented and found error introduced. Hence, we have correctedit accordingly. Please see a clean version of manuscript page number 4-line number 41-42

3. In the description of the study area please do mention the private clinics as well. Also, do add basic description of coverage of institutional deliveries and antenatal care.

Response 3: Thank you, we have accepted the comments, and considered in the study setting section, page 6 line 91.

4. On sampling procedures please use words to explain what is that k-interval for the systematic sampling. See further comments of the reviewer below.

Response 4: We have accepted all the comments, and corrected all, see clean version of manuscript page 7 line 111.

5. The operational definition section needs a table with the list of things to make these definitions. It would be beneficial if the English instrument were added in the supplements.

Response 5: Thank you, we have accepted, and summarized in table 1, please see under the operational definition, page 8. 

6. The data processing and analysis section need to be expanded.

Response 6: Thank you for the comments, we entertained in the data processing and analysis section.

7. We do not analyze a logistic regression. We use logistic regression to analyze something. Please correct.

Response 7: We authors forwarded great thanks; we have addressed it.

8. There is use of confidence intervals for a proportion in the results. How such thing interval was estimated? This is something to put in the methods

 Response 8: We have corrected it, see under data process and analysis section.

9. Line 219 is for the “data processing and analysis” section not in the “results”. Please correct.

 Response 9: We have corrected it.

10. When report the mean as in the line 163 we usually write the SD as, for example, “(SD = 5.76)”. Notice the equals sign

 Response 10: It is corrected based on comment, see page 11 line 162

11. Remove the “The” on “The proportion of mistreatment during facility-based childbirth”

Response 11: We have removed it.

12. Please report the confidence intervals for all mistreatment types in text

Response 12: We authors accepted the comment and we have included confidence interval for all mistreatment types.

13. Comments regarding table 1; 

a) Add the mean and SD of the age, 

b) Make all percentages 1 decimal place, 

c) Add more categories for the “family estimated monthly income” variable. and please provide quartiles.

d) On “HCP conducting ANC follow up” variable please correct the “Privet clinic”. That should be “Private clinic”. 

e) Please check the alignment for “complication during delivery”.

Response 13: All the comments are addressed accordingly, see on page 16, table 3. 

14. Comments regarding table 3;

a) for the ORs please make all 2 decimal places, make sure that references have 1 (for example it is missing on Educational status)

Response 14 a. All are corrected based on your comments

b) Why age and “religion” are not included? Regardless of the p-value. This is a key variable.

Response 14 b: Thank you for your view, we have included both variables in the revised manuscript, please see in the table 4.

15. Comments regarding figure 1

a) This pie is taking a lot of space to inform exactly the same in text. And it does not provide any confidence interval as in the text you did. Drop it. I suggest to add this information on figure 2.

Response 15: We haveaccepted the suggestion and corrected accordingly

16. Comments regarding figure 2

a) Add the proportion of mistreatment you have on pie. Add this to the bar plot.

b) Add to all of these bar error bars to capture the confidence interval

c) Fix the decimal places to 1

d) Remove the percentage symbol in the plot

e) Remove the so called main title this can be placed in the x-axis and in caption of the plot

 Response 16: We have accepted all the comments and correction are made accordingly

17. Discussion

a) Lines 253 to 257. The reason for the differences did you check that these studies define the same way as you do here? 

Response 17a: The studies define mistreatment in different ways, for example study conducted in Addis Ababa to define mistreatment were used six categories while in current study used seven categories disrespect and abuse. 

b) Are the populations comparable besides the health facility? 

Response 17b: 

Beside the study participants, there is variation in proportion of previous history of institutional delivery, in current study 21% of the study participants had no previous history of institutional delivery, in studies conducted Addis Ababa (31%), and Arba Minch (37%). The absence of pervious history of institutional delivery was reported as risk factor for childbirth care mistreatment in current study. 

Thank you, dear editor, for suggestion, we have accepted the suggestion and included under discussion section.

c) Be very careful. We do not want to label medical schools as the center of mistreatment

Response 17c: We apologized for interpretation error. we have accepted the comment and corrected it.

d) What the authors thing of the tool they used to assess the mistreatment? Did you check for validity to use in this setting? Please discuss this. And most likely this must be considered for the limitation subsection.

Response 17d: In this study, mistreatment measurement tool was developed from Maternal and Child Health Integrated Program (MCHIP) respectful maternity care tool kit which was not validated in Ethiopia. We have accepted the comment and acknowledge as limitation in limitation section.

e) The strengths and limitations need more elaboration. This is a health facility-based study therefore a potential for selection bias may doom the generalization of this study. There is always possibility of confounding this must be discussed.

Response 17e: It is corrected based on comments

18. Please fill the STROBE statement and add to the supplements

Response 18: We have filled the STROBE statement and submitted in supplementary file list.

19. Add a specific and clear section of ethics

 Response 19: We have includedethics under method section 

20. Comments regarding abstract: -

a) OR please make sure they have 2 decimal places

b) For all mistreatment proportion make sure they have confidence intervals

 Response 20: We have accepted and have corrected all the comments accordingly 

21. Please provide additional details regarding participant consent. In the ethics statement in the Methods and online submission information, please ensure that you have specified what type you obtained (for instance, written or verbal, and if verbal, how it was documented and witnessed). If your study included minors, state whether you obtained consent from parents or guardians. If the need for consent was waived by the ethics committee, please include this information.

Response 21: We have included the comments in ethics statement section. Consent from parents or guardians was not taken because all the study participants have already established their own family that even though few participants are aged below 18 years, they are mature minors. Hence, we did not expected to get consent from their family. 

22. In your Methods section, please provide additional information about the participant recruitment method in particular a description of how participants were recruited.

Response 22: We have accepted the description about sampling procedure was shallow in the first submission. In this revised manuscript we have entertained the sampling procedure in detail. 

23. You indicated that you had ethical approval for your study. In your Methods section, please ensure you have also stated whether you obtained consent from parents or guardians of the minors included in the study or whether the research ethics committee or IRB specifically waived the need for their consent

Response 23: We have included the comments in ethics statement section. Consent from parents or guardians was not taken because all the study participants have already established their own family that even though few participants are aged below 18 years, they are mature minors. Hence, we did not expect to get consent from their family.

Moreover, according to the Ethiopian national research ethics review guideline, consent from parents or guardians for adolescents aged less than 18 years is waivered for some cases. Among these emancipated/mature minors, one is, those who married or established their own family. Hence, consent was not obtained from parents or guardians for these pregnant adolescents. This information is noted in the main manuscript with citation of the reference.

24. We suggest you thoroughly copyedit your manuscript for language usage, spelling, and grammar.

 Response 24: Thank you. We tried to edit the language.

25. During our internal evaluation of the manuscript, we found significant text overlap between your submission and the following previously published works:

- https://www.biorxiv.org/content/10.1101/430199v1(Introduction, paragraph 1, sentences 3-5) (Introduction, paragraph 4, sentences 1-2) (Discussion, paragraph 13, sentence 3)

- https://www.tandfonline.com/doi/full/10.1080/09688080.2018.1502020 (Introduction, paragraph 2, sentence 1)

- https://www.researchsquare.com/article/rs-107805/v1(Introduction, paragraph 2, sentence 2)

- https://preview-reproductive-healthjournal.biomedcentral.com/articles/10.1186/s12978-015-0024-9 (Types of mistreatment during facility-based childbirth, sentences 1-2) (Discussion, paragraph 5, sentence 4)

-https://journals.plos.org/plosone/article?id=10.1371%2Fjournal.pone.0205545(Discussion, paragraph 2, sentence 4) (Discussion, paragraph 12, sentence 4)

- https://obgyn.onlinelibrary.wiley.com/doi/abs/10.1016/j.ijgo.2014.08.015 (Discussion, paragraph 6, sentence 4)

- https://bmcpregnancychildbirth.biomedcentral.com/articles/10.1186/s12884-018-1970-3 (Discussion, paragraph 11, sentence 4)

 Response 25: We accepted the all comments and we made correction 

26. Provide details as to how the current manuscript advances on previous work.

 Response 26: We authors believed that this research work adds the following body of knowledge with the existed evidence:

• Evidence from the previous studies focused on hospital-level settings, while the current study involved nine health centers, where disrespect and abuse occur.

• In our study non-confidentiality care is the most experienced form of abuse and respect during health facility childbirth, which is not reported by the previous studies.

• A new factorwas identified by this study such as having complication during childbirth and lack of or less than four antenatal care visits was associated with mistreatment.

Reviewer 1'

1. Grammar and spelling: The paper is written in poor English, has multiple grammatical and spelling errors throughout the entire document. I have highlighted most of them in the manuscript. Recommendation: the authors need to rewrite the paper in standard English to enable the readers to understand.

Response 1: Thank you. We tried to edit the language.

2. Justification (line 72-74): the two statements contradict each other. Which is the correct position?

Response 2: Thank you dear reviewer to your comment, we have rewritten the whole paragraph. The lack of information regarding disrespect and abuse in the study area is the correct statement. 

3. Study site: 

a) it is not clear whether the study was conducted in one or multiple study sites. 

Response 2a: Dear reviewer, in this study eight health centers and one compressive specialized hospital were included which means multiple study sites were considered

b) What is the name of the facility? 

Response 3b: In this study, all childbirth care providing health facilities; eight health centers, and one referral hospital were considered. Gondar University compressive specialized hospital, Azezo health center, Gondar health center, Merak health center, Woleka health center, Teda health center, Genbot 20 health center, Biajing health center, and Gebreale health center were included in the study. For further information, We haveincluded it under sampling procedure section.

c) How was it chosen from among the other 14 facilities in the city?

Response 3c: In this study, all childbirth care providing health facilities in city; eight health centers, and one referral hospital were considered. The number 14 in study setting section indicate that number of health posts. Health posts currently not provide delivery care (childbirth care).

d) What are the numbers of deliveries handled by this facility per annum? 

Response 3d: The average number of deliveries per month in health institutions was 1,173 (Gondar University comprehensive specialized hospital, Azezo health center, Gondar health center, Merak health center, Woleka health center, Teda health center, Genbot 20 health center, Biajing health center, and Gebreale health center had averagely 928, 66, 38, 40, 25, 50, 11,7,8 skilled births per month respectively) (10). The total coverage of institutional deliveries and antenatal care was 62%, and 86% respectively. The number of deliveries of Gondar University comprehensive specialized hospital is high when compared to other facilities, this might be because of referral hospital. We have mention it under study setting and sampling procedure sections.

e) What made this site the most ideal for your study?

Response 3e: the reason to select this site 

The first reason is referral site, Gondar University comprehensive specialized hospital handle high number of deliveries per month (928), high number of client flow might be affecting the quality of care. 

The second reason, in this study site there is high discrepancy between in skill deliveries and antenatal carevisits, it also indicates low quality of care. 

Third reason, in this study area the number of deliveries handledby health centers was low.

4. Sample size: the explanation after sample size calculation is not clear. recommendation: edit for clarity

 Response 4: Thank you, we have edited for clarity

5. Comments regarding data collection procedures: 

a) Review the whole of this section for consistency in grammar eg the use of "is" instead of "was".

 Response 5a: Thank you, we have corrected rigorously

b) Clarify the role of the degree nurse’s vis a vis the principal investigator

Response 5b: The principal investigator oversees all activities of supervisors (degree nurses) and data collectors, whereas supervisors (degree nurses) did supervision of all the data collectors’ activities.

c) What research experience did the six research assistants have?

d) Response 5c: For data collection, we are employed experienced data collectors in data collection and supervision activities from Tseda health science college staffs. These data collectors were priorly participated in different projects in data collection and supervision activities.

e) Were the research assistants employees of the same hospitals or were they independently employed for the study only? how did you address the issue of bias if these nurses worked in the same facilities where they conducted the interviews?

Response 5e: Our study is free from reporting bias because we were independently employed data collectors for data collection from Tseda health science college.

f) Lots of redundancies and repetitions eg questionnaire was pretested (written more than twice)

 Response 5f: We authors accepted and have corrected it

6. Comments regarding results: 

a) clarify if n=574 or 584(all tables). 

Response 6a: Our study, 584 estimated sample size while 574 the number of response (n=574). it was editorial problem; We have corrected it.

b) Table 2: In some of the cells eg previous ANC attendance, the "n" is not 574 as some of the women were carrying a first pregnancy

Response 6b: Thank you, this was typing error from SPSS output in to the table. We have corrected it, please see from table 3.

7. Discussion: some of the explanations are not coherent e.g. why pregnant women who delivered in a hospital before were likely to be mistreated. Review the explanations

 Response 7: We authors accepted the comment given and corrected it

8. Strengths and limitations: too brief and not clear

 Response 8: We have rewrite and tried to make it clear.

9. Conclusions: Edit for clarity

 Response 9: We have made editorial correction 

10. Ethics and consent: did the study pose any risks to the study participants? what care was offered to the abused women? were they linked to any care?

 Response 10:

Participating in this study did not pose the study participants to any risk because first, the interview were exit interview; second, the confidentiality and privacy were highly secured. 

At spot of data collection, determining (knowing) abused woman is difficult, because disrespect and abuse measurement is a composite measurement and it need SPSS analysis to identify abused women. However, after analysis and report write up, we present our findings to health facilities health care manager and health sectors to design intervention. 

11. Did you obtain parental consent for those below 18 years of age. 

Response 11: We didn't obtain consent from respondents' mother because all the respondents establish their own family and gave birth. Hence, though few respondents' age was between 15 years and 18 years old, these women were treated as mature minors that parental consent is not needed. Their consent is enough. Moreover, according to the Ethiopian national research ethics review guideline, consent from parents or guardians for adolescents aged less than 18 years is waivered for some cases. Among these emancipated/mature minors, one is, those who married or established their own family. Hence, consent was not obtained from parents or guardians for these pregnant adolescents. This information is noted in the main manuscript with citation of the reference. 

12. Edit the grammar for clarity

 Response 12: Thank you. We tried to edit the language.

---

## [Decision Letter · Decision Letter 1]

5 Aug 2021

PONE-D-21-04196R1

High attribution of non-consented care for high prevalence of mistreatment among postpartum mothers who give birth at public health facilities in Gondar city, northwest Ethiopia

PLOS ONE

Dear Dr. Boke,

Thank you for submitting your manuscript to PLOS ONE. After careful consideration, we feel that it has merit but does not fully meet PLOS ONE’s publication criteria as it currently stands. Therefore, we invite you to submit a revised version of the manuscript that addresses the points raised during the review process.

We look forward to receiving your revised manuscript.

Kind regards,

Orvalho Augusto, MD, MPH

Academic Editor

PLOS ONE

Journal Requirements:

Additional Editor Comments (if provided):

This is the second revision of this report on the prevalence of mistreatment among postpartum mothers. The authors did respond to many of the issues raised in the first revision. However, there are still issues and new orthographic mistakes were introduced.

1. Introduction

- Line 46 - maternal mortality ratio of 70,000 per 100,000 live births?

- Line 47 - Why this “please check the number 70,000”.

- Line 58 - ”A laboring mother is subjected toa diverse”. Please correct the “toa”

- Line 61 - put a space before “Mistreatment”

- Lines 70 to 74 there are a lot of strange interval mistakes. I cannot understand why the percentages are repeated.

2. Methods

- Line 89 - the “KM” should be “Km”

- Line 94 - what is this “2011 EFY”? Why not a more recent reference?

- Line 104 - remove this “Finally, the largest sample size was used”

- Line 109 - put space after “.” In the whole document, there is a lot of these.

- One thing I cannot get from this report is where the women were recruited from within the public health facilities. This is an important detail.

3. Results

- Line 172 - put space after 17.2%.

- Page 17 - when reporting the prevalence, as for example the one in line 201, please write, for example, 171 (29.8%, 95% CI: 25.4 to 32.8%). Revise all proportions on this page including the one in line 197.

- Please report the odds ratio and their confidence intervals with 2 decimal places. This applies to the abstract, page 18 and table 4.

- Line 222 remove the “at P-value of loss than 0.05”. Do not resume associations to a matter of p-values.

4. Discussion

- Please revise spaces like line 158 after “.”

Reviewers' comments:

Reviewer's Responses to Questions

**Comments to the Author**

1. If the authors have adequately addressed your comments raised in a previous round of review and you feel that this manuscript is now acceptable for publication, you may indicate that here to bypass the “Comments to the Author” section, enter your conflict of interest statement in the “Confidential to Editor” section, and submit your "Accept" recommendation.

Reviewer #1: All comments have been addressed

Reviewer #2: (No Response)

2. Is the manuscript technically sound, and do the data support the conclusions?

Reviewer #1: Yes

Reviewer #2: Yes

3. Has the statistical analysis been performed appropriately and rigorously? 

Reviewer #1: Yes

Reviewer #2: No

4. Have the authors made all data underlying the findings in their manuscript fully available?

Reviewer #1: No

Reviewer #2: No

5. Is the manuscript presented in an intelligible fashion and written in standard English?

Reviewer #1: Yes

Reviewer #2: No

6. Review Comments to the Author

Reviewer #1: I have gone through my previous comments to the authors of the paper, the author response and both the manuscript with track changes and the final copy, and, I am satisfied with the author's response to my comments.

Reviewer #2: Title:

I would like to suggest a change in title. What do you think about “prevalence and risk factor for mistreatment in childbirth: a survey in a Gondar City, Ethiopia.

I think easier to the reader.

Intruduction

Line 43 - Define the initials: Maternal Mortality Ratio (MMR)

Line 48 – it’s not 70,000; it is 70.

Line 55 – this - small letter

Line 57 and line 72 – disrespect and abuse have different definition from mistreatment. (Bohren et al., 2015; Bowser & Hill, 2010)

Line 55 – 61 – define mistreatment in childbirth or disrespect and abuse. (Bohren et al., 2015; Bowser & Hill, 2010)

Line 69 - three out of four/ three in four

Line 70 - four hundred twenty mothers out of / in one hundred thousand live 71 birth

Coments:

The first paragraph - I think the first paragraph is important to explain Africa and Ethiopia's context. However, this paragraph is not about the major issue (mistreatment). I suggest starting the manuscripts by talking about mistreatment and put information about maternal mortality throughout the introduction.

The introduction talking about mistreatment as a strategy to prevent maternal mortality. However, mistreatment should cause death in a small part of victims. It’s mean that to prevent maternal mortality other issues should be addressed before as antenatal care and health assistance in labor/childbirth, for example.

Mitigate mistreatment is important to improve women's human rights and decrease the risk of consequences as postpartum depression and evasion of health services (Leite et al., 2021; Leite, Pereira, Leal, & Silva, 2020; Silveira; et al., 2019). Another possibility is explaining better that women have afraid to go to the hospital due to mistreatment and this behavior puts women at risk to die.

What are the risk factors to suffer mistreatment in Ethiopia or Africa? Nothing about this was informed.

Methods

In “population” is not clear the hospital (s) participants. The research was conducted in one hospital because there is only one hospital in the city? Or there were other Hospitals/health services? I think this situation could be clearest in the text.

Manuscript: In “Population” – “All postpartum mothers who gave birth at public health facilities of Gondar city during a period 93 of data collection were considered as the study population.”

Manuscript: In “Sample size determination and Sampling Procedure” – “Finally, to get the study 100 participants systematic sampling method with k-interval of two were employed at each facility”

I cannot understand the sampling. It was the census of all women in postpartum between March and April 2019 or it was a sampling of all eligible women.

The definition of mistreatment is not according to the mistreatment definition used by WHO. The definition used is about “disrespect and abuse”.

Line 102 – 127 - The authors could to presents the original question which was asked for women

The topic “Operational definition” should be before “Data Collection Tool and procedures”

The line “That is, a total 136 of 24 verification criteria of mistreatment were included.” Should be in topic “Operational definition”

The topic “Operational definition” should describe all variables included in the analysis.

There no information about Ethics committee approval

Results

Table 1 – income: inform the currency

Line 218 – 220 – I suggest change this sentence to “statistical Analysis”

Comments: Commonly, the choices about variables that compose the multivariate logistic regression have been done based on statistical criteria. But, in my point of view, this choice should be based on in theoretical model using the DAG definition about causality and confounders.

Based on my own country, type of birth and parity are important predictors of mistreatment in labor. I suggest adding this information to the model.

history of previous institutional delivery is a complicated variable. This should be used only if a woman is multiparous. To primiparous women, this question is not applicable. Therefore, this variable just could be used in a sensitivity analysis with multiparous women.

Suggests: I suggest conducted some multivariate logistic regression stratifying by parity and type of birth.

Discussion:

In my opinion, the discussion about prevalence should offer more than a simple comparison between the current study and the other studies. The context of the hospitals should be considering and why health care providers use to mistreat women in Ethiopia. The prevalence was very high.

Considering the discussion about risk factors, the authors could explain why (the causal mechanism) some variables present as a risk factor. Comparison with other studies is important to observe some consistency of the findings. However, the interpretation of the finding should be the major topic.

Strength and limitation

I have some doubts if collect information immediately following delivery and inside the hospital/health center is a good option to measure mistreatment. Some women could have been afraid to address this issue and suffer consequences (more mistreatment) due to these acts. A second point is some women could present the “gratitude bias” to be alive and with their baby in arms. Thereby, only a few days after the delivery they get realize all violence and mistreatment that she has suffered.

Do not address fundal pressure and episiotomy could be a limitation

Conclusion

Knowing the mistreatment prevalence is high, what action it is possible to take aiming to mitigate mistreatment in Ethiopia?

References:

Bohren, M. A., Vogel, J. P., Hunter, E. C., Lutsiv, O., Makh, S. K., Souza, J. P., … Gülmezoglu, A. M. (2015). The Mistreatment of Women during Childbirth in Health Facilities Globally: A Mixed-Methods Systematic Review. PLoS Medicine, 12(6), 1–32. https://doi.org/10.1371/journal.pmed.1001847

Bowser, D., & Hill, K. (2010). Exploring Evidence for Disrespect and Abuse in Facility-Based Childbirth Report of a Landscape Analysis. Harvard School of Public Health University Research Co., LLC. https://doi.org/10.1624/105812410X514413

Leite, T. H., Gomes, T., Marques, E. S., Pereira, A. P. E., Silva, A. A. M. da, Nakamura-Pereira, M., & Do Carmo Leal, M. (2021). Association Between Mistreatment of Women during Childbirth and Postnatal Maternal and Child Health Care: Findings from “Birth in Brazil.” Women and Birth.

Leite, T. H., Pereira, A. P. E., Leal, M. do C., & Silva, A. A. M. da. (2020). Disrespect and abuse towards women during childbirth and postpartum depression: findings from Birth in Brazil Study. Journal of Affective Disorders, 273, 391–401.

Silveira;, M. F., Mesenburg;, M. A., Bertoldi;, A. D., Mola;, C. L. De, Bassani;, D. G., Domingueses;, M. R., … Coll, C. V. N. (2019). The association between disrespect and abuse of women during childbirth and postpartum depression: Findings from the 2015 Pelotas birth cohort study. Journal of Affective Disorders, 256(April), 441–447. https://doi.org/10.1016/j.jad.2019.06.016

7. PLOS authors have the option to publish the peer review history of their article (what does this mean?). If published, this will include your full peer review and any attached files.

Reviewer #1: No

Reviewer #2: **Yes: **Tatiana Henriques Leite

---

## [Author Response · Author response to Decision Letter 1]

27 Aug 2021

To: Editor-in-chief of PLOS ONE Journal 

Subject: Submission of a revised manuscript for publication

Dear Editor,

Thank you for allowing us to submit a revised draft of our manuscript entitled “Prevalence and risk factor for mistreatment in childbirth: in Gondar city health facilities, Ethiopia "[Manuscript ID number: PONE-D-21-04196]. We appreciate the time and effort dedicated by you and the reviewers to provide your valuable feedback on our manuscript. We are grateful to the reviewers for their insightful comments which improve our manuscript.

We accepted and tried to incorporate all of the comments provided. Thus, the comments are attached here below with their point-by-point responses given in blue font color. Besides, the detailed changes made are highlighted in the “revised manuscript with track changes” to easily identify the changes/improvements, and clean copy of the revised manuscript are prepared. 

Response to Editor comments 

Introduction

1. Line 46 - maternal mortality ratio of 70,000 per 100,000 live births? Line 47 - Why this “please check the number 70,000”.

Response: Thank dear editor for comments, this is typing error, we apologies to error. We have corrected it on revised version of manuscript. The correct statement is “maternal mortality ratio 70 per 100, 0000 live birth”.

2. - Line 58 -” A laboring mother is subjected toa diverse”. Please correct the “toa”

- Line 61 - put a space before “Mistreatment”

- Lines 70 to 74 there are a lot of strange interval mistakes.

- I cannot understand why the percentages are repeated.

Response: we authors accepted all the comments and corrected accordingly.

Methods

1. Line 89 - the “KM” should be “Km”

Response: Thank for your comment, we have corrected 

2. Line 94 - what is this “2011 EFY”? Why not a more recent reference?

Response: Thank you editor for clarification question, EFY means Ethiopian finical year, 2011 EFY is equivalent to 2019 G.C., on the revised document we have included Gregorian calendar date. 

3. Line 104 - remove this “Finally, the largest sample size was used”

Response: Thank for your comment, we have corrected it 

4. Line 109 - put space after “.” In the whole document, there is a lot of these.

Response: Thank for your comment, we have corrected it 

5. One thing I cannot get from this report is where the women were recruited from within the public health facilities. This is an important detail.

Response: Thank you editor for clarification question, we were recruited study participants at health before exit. This might be expose to bias, we have tried to acknowledge this bias as limitation in limitation section. 

 Results

Line 172 - put space after 17.2%. 

Page 17 - when reporting the prevalence, as for example the one in line 201, please write, for example, 171 (29.8%, 95% CI: 25.4 to 32.8%). Revise all proportions on this page including the one in line 197.

Please report the odds ratio and their confidence intervals with 2 decimal places. This applies to the abstract, page 18 and table 4.

Line 222 remove the “at P-value of loss than 0.05”. Do not resume associations to a matter of p-values.

Response: we authors accepted all the comments and corrected accordingly, thank you.

Discussion

1. Please revise spaces like line 158 after “.”

Response: Thank you dear Editor for your comment, we have corrected the comment.  

Response to Reviewer #2 comments

1. I would like to suggest a change in title. What do you think about “prevalence and risk factor for mistreatment in childbirth: a survey in a Gondar City, Ethiopia. I think easier to the reader.

Response: Thank you reviewer for comment. We believe that title is an advertising of the work. As a result, we wrote as “high attribution of non-consented care for high prevalence of mistreatment among postpartum mothers who give birth at public health facilities: in Gondar city, northwest Ethiopia” However, to make easy to reader we modify as “Prevalence and risk factor for mistreatment in childbirth: in Gondar city health facilities, Ethiopia”.

Introduction 

2. Line 43 - Define the initials: Maternal Mortality Ratio (MMR)

Response: Thank you reviewer for comments, we have incorporated the comments under introduction section line 47-49. 

3. Line 48 – it’s not 70,000; it is 70.

Line 55 – this - small letter

Response: Thank you dear reviewer for comments, this is typing error, we apologize to error. We have corrected on revised version of manuscript. The correct statement is “maternal mortality ratio 70 per 100, 0000 live birth”.

4. Line 57 and line 72 – disrespect and abuse have different definition from mistreatment. (Bohren et al., 2015; Bowser & Hill, 2010), Line 55 – 61 – define mistreatment in childbirth or disrespect and abuse. (Bohren et al., 2015; Bowser & Hill, 2010)

Response: Thank you reviewer for constructive comments, we define mistreatment under introduction section line 39-41. 

5. Line 69 - three out of four/ three in four

Line 70 - four hundred twenty mothers out of / in one hundred thousand live 71 birth

Response: Thank you for suggestion, we have accepted comments and corrected. 

6. The first paragraph - I think the first paragraph is important to explain Africa and Ethiopia's context. However, this paragraph is not about the major issue (mistreatment). I suggest starting the manuscripts by talking about mistreatment and put information about maternal mortality throughout the introduction.

Response: Thank you reviewer for your valuable comments, we have tried to incorporate all comments, please see first paragraph under introduction section. 

7. The introduction talking about mistreatment as a strategy to prevent maternal mortality. However, mistreatment should cause death in a small part of victims. It’s mean that to prevent maternal mortality other issues should be addressed before as antenatal care and health assistance in labor/childbirth, for example. Mitigate mistreatment is important to improve women's human rights and decrease the risk of consequences as postpartum depression and evasion of health services (Leite et al., 2021; Leite, Pereira, Leal, & Silva, 2020; Silveira; et al., 2019). Another possibility is explaining better that women have afraid to go to the hospital due to mistreatment and this behavior puts women at risk to die.

Response: Thank you reviewer for your valuable comments, we have made modification, see line 74-75 

8. What are the risk factors to suffer mistreatment in Ethiopia or Africa? Nothing about this was informed.

Response: Thank you for your comments, we have summarized risk factors to mistreatment in Africa under line 86-88.

Methods section 

1. In “population” is not clear the hospital (s) participants. The research was conducted in one hospital because there is only one hospital in the city? Or there were other Hospitals/health services? I think this situation could be clearest in the text. Manuscript: In “Population” – “All postpartum mothers who gave birth at public health facilities of Gondar city during a period 93 of data collection were considered as the study population.”

Response: Thank you dear reviewer for your clarification question, in Gondar city nine health facilities (one hospital and eight health centers) provide delivery service, in our study all delivery service providing health facilities were included. 

2. Manuscript: In “Sample size determination and Sampling Procedure” – “Finally, to get the study 100 participants systematic sampling method with k-interval of two were employed at each facility” I cannot understand the sampling. It was the census of all women in postpartum between March and April 2019 or it was a sampling of all eligible women.

Response: Thank you again for your clarification question. To get study participants from each delivery service providing health facilities. First, the estimated sample size was proportionally allocated to health facilities (the eight health centers and one referral hospital) based on their average monthly deliveries number. Then, finally, to get study participants (allocated sample size) from each facility a systematic sampling technique with two k intervals employed, until required sample size attains.

3. Line 102 – 127 - The authors could to presents the original question which was asked for women

Response: Thank you dear reviewer for your comments, during first review the editor and first reviewer request us to supplement the questionnaire, as per request we had already submitted the questionnaire as supportive file. 

4. The topic “Operational definition” should be before “Data Collection Tool and procedures” That is, a total 136 of 24 verification criteria of mistreatment were included.” Should be in topic “Operational definition”

Response: Thank for comments, we have corrected the comments accordingly. 

5. The line the topic “Operational definition” should describe all variables included in the analysis.

Response: Thank you dear reviewer for your comment, we have described all variables under operational definition section, see line 130-140 

6. There no information about ethics committee approval

Response: Thank you reviewer for your suggestion, we have already state about ethics under methods section, page 11 line 175-187

Results section 

1. Table 1 – income: inform the currency

Line 218 – 220 – I suggest change this sentence to “statistical Analysis”

Response: Thank you reviewer for your comments, we have corrected it.

2. Comments: Commonly, the choices about variables that compose the multivariate logistic regression have been done based on statistical criteria. But, in my point of view, this choice should be based on in theoretical model using the DAG definition about causality and confounders.

 Response: Thank you for comments, we added variables in multivariate logistic regression, based on their statistical criteria, in additional to this, significantly associated variables in previous studies were included in multivariate logistic regression analysis without considering statistical criteria. 

3. Based on my own country, type of birth and parity are important predictors of mistreatment in labor. I suggest adding this information to the model.

Response: Thank dear reviewer for your suggestion, we have added both variables (type of birth and parity) in the model irrespective of their p-value. 

4. history of previous institutional delivery is a complicated variable. This should be used only if a woman is multiparous. To primiparous women, this question is not applicable. Therefore, this variable just could be used in a sensitivity analysis with multiparous women.

Response: Thank you for your comments, we have removed complicated variable (history of previous institutional delivery) from model to avoid effect and we did reanalysis. 

5. Suggests: I suggest conducted some multivariate logistic regression stratifying by parity and type of birth.

Response: Thank you for your valuable comments, we have done some logistic regression stratification analysis by parity and type of birth, please see, line 251-261, table 5 &6 in revised version of manuscript. 

Discussion:

1. In my opinion, the discussion about prevalence should offer more than a simple comparison between the current study and the other studies. The context of the hospitals should be considering and why health care providers use to mistreat women in Ethiopia. The prevalence was very high.

Response: thank you for your comments, as I early mentioned, our study is more focused on evaluating health care provider and women behavior related mistreatment dimensions, hospital and health facility environment constrain were not assessed, due to this reason we can’t support our finding with health system and facility environment related reasons. We accepted it as limitation 

2. Considering the discussion about risk factors, the authors could explain why (the causal mechanism) some variables present as a risk factor. Comparison with other studies is important to observe some consistency of the findings. However, the interpretation of the finding should be the major topic.

Response: Thank you for comments, we have tried to explain causal mechanism for some risk factors, please see discussion section.

Strength and limitation

1. I have some doubts if collect information immediately following delivery and inside the hospital/health center is a good option to measure mistreatment. Some women could have been afraid to address this issue and suffer consequences (more mistreatment) due to these acts. A second point is some women could present the “gratitude bias” to be alive and with their baby in arms. Thereby, only a few days after the delivery they get realize all violence and mistreatment that she has suffered. Do not address fundal pressure and episiotomy could be a limitation.

Response: Thank your dear reviewer for your constrictive comments, we have accepted all the comments and we have tried to acknowledged all these limitation under limitation section. 

Conclusion

1. Knowing the mistreatment prevalence is high, what action it is possible to take aiming to mitigate mistreatment in Ethiopia?

Response: Thank you reviewer for comments, to alleviate mistreatment health mangers need to strengthening actions, like providing maternity education during antenatal care and appropriate management of complications to improve the quality of maternity care at health facilities. Also, health care providers that work in childbirth care need to be trained on the importance of informed consent and give compassionate and respectful care. Moreover, Health facilities need to promote positive birth experiences through the provision of respectful, dignified, supportive, and consented care. We state this recommendation under conclusion section. 

 

 References:

Bohren, M. A., Vogel, J. P., Hunter, E. C., Lutsiv, O., Makh, S. K., Souza, J. P., … Gülmezoglu, A. M. (2015). The Mistreatment of Women during Childbirth in Health Facilities Globally: A Mixed-Methods Systematic Review. PLoS Medicine, 12(6), 1–32. https://doi.org/10.1371/journal.pmed.1001847

Bowser, D., & Hill, K. (2010). Exploring Evidence for Disrespect and Abuse in Facility-Based Childbirth Report of a Landscape Analysis. Harvard School of Public Health University Research Co., LLC. https://doi.org/10.1624/105812410X514413

Leite, T. H., Gomes, T., Marques, E. S., Pereira, A. P. E., Silva, A. A. M. da, Nakamura-Pereira, M., & Do Carmo Leal, M. (2021). Association Between Mistreatment of Women during Childbirth and Postnatal Maternal and Child Health Care: Findings from “Birth in Brazil.” Women and Birth.

Leite, T. H., Pereira, A. P. E., Leal, M. do C., & Silva, A. A. M. da. (2020). Disrespect and abuse towards women during childbirth and postpartum depression: findings from Birth in Brazil Study. Journal of Affective Disorders, 273, 391–401.

Silveira;, M. F., Mesenburg;, M. A., Bertoldi;, A. D., Mola;, C. L. De, Bassani;, D. G., Domingueses;, M. R., … Coll, C. V. N. (2019). The association between disrespect and abuse of women during childbirth and postpartum depression: Findings from the 2015 Pelotas birth cohort study. Journal of Affective Disorders, 256(April), 441–447. https://doi.org/10.1016/j.jad.2019.06.016

Response: Thank you dear reviewer for your references suggestion, the references are too helpful to improve our manuscript, we used all references.

---

## [Decision Letter · Decision Letter 2]

15 Feb 2022

PONE-D-21-04196R2Prevalence and risk factor for mistreatment in childbirth: in Gondar city health facilities, EthiopiaPLOS ONE

Dear Dr. Boke,

Thank you for submitting your manuscript to PLOS ONE. After careful consideration, we feel that it has merit but does not fully meet PLOS ONE’s publication criteria as it currently stands. Therefore, we invite you to submit a revised version of the manuscript that addresses the points raised during the review process.

We look forward to receiving your revised manuscript.

Kind regards,

Orvalho Augusto, MD, MPH

Academic Editor

PLOS ONE

Journal Requirements:

Reviewers' comments:

Reviewer's Responses to Questions

**Comments to the Author**

1. If the authors have adequately addressed your comments raised in a previous round of review and you feel that this manuscript is now acceptable for publication, you may indicate that here to bypass the “Comments to the Author” section, enter your conflict of interest statement in the “Confidential to Editor” section, and submit your "Accept" recommendation.

Reviewer #3: (No Response)

2. Is the manuscript technically sound, and do the data support the conclusions?

Reviewer #3: Yes

3. Has the statistical analysis been performed appropriately and rigorously? 

Reviewer #3: I Don't Know

4. Have the authors made all data underlying the findings in their manuscript fully available?

Reviewer #3: (No Response)

5. Is the manuscript presented in an intelligible fashion and written in standard English?

Reviewer #3: No

6. Review Comments to the Author

Reviewer #3: Thank you for the opportunity to review your paper. I appreciate all of the effort that the authors put into submitting and then revising your manuscript.

I have a few comments and edits for your consideration:

1.) Suggest editing title to “Prevalence and risk factors for mistreatment in childbirth: in health facilities in Gondar City, Health facilities, Ethiopia

2.) Abstract-Background section:

a. Is compassionate RMC ensuring women’s survival? I’m unsure of this statement although I do understand that more positive experiences of care could lead to better health outcomes for women and newborns (based on improving the coverage of facility deliveries).

b. You say that compassionate and respectful maternity care has received less attention both in practice and research in Ethiopia I think there is quite a bit of literature about RMC in Ethiopia. Perhaps there is less data from Gondar City specifically and if so, I might state that.

3.) Abstract—Methods

a. How many facilities was this study conducted in?

b. These are postpartum women—how long postpartum?

c. You say: A binary logistic regression analysis was done to see the association between mistreatment and the independent variables. Suggestion: A binary logistic regression analysis was done to see whether there was an association between mistreatment and independent variables such as XXXXXX

d. You say: Finally, the logistic regression analysis was done by stratifying type of party and mode of delivery. I think you mean parity?

4.) Abstract—Results

a. Suggest changing the word “Result” to “Results” on line 27

b. Line 31—change “complication” to “complications” on line 31

c. Did you adjust for facility-based clustering?

5.) Abstract—Conclusions

a. Suggest capitalizing the word “the” in line 34

b. The conclusion that the proportion of women who were mistreated was higher here than in other developing countries—it seems like it would be useful to compare what you found with other studies in Ethiopia since there have been quite a few published

c. You mention some possible ways to address mistreatment based on your findings. How would you address the issue that mistreatment was more prevalent in hospitals?

d. In your conclusion you state that the proportion of mistreatment was higher in this study but the in the introduction of the paper, you do show ranges of up to 98.9%

Introduction

• Line 43: Maternal mistreatment at childbirth is more than just what you’ve mentioned here. I suggest using a more accepted definition of RMC

Methods

• Line 152: Data were collected through face-to-face interviews. Were these interviews conducted in the health facilities? A chart review was also done but which data/indicators were extracted from the charts?

Results

• Line 203—The majority of participants had ANC visit—I assume this is at least one ANC visit?

• Line 245—You state that mothers who had less than 4 ANC visits were 3.58 times more likely to mistreat—but should be changed to experience mistreatment rather than mistreat; this is the same for the next few sentences—the women themselves are not mistreating but are rather being mistreated

• Line 251—"The analysis was stratified by party…”; this should be parity

Conclusions:

• Line 277—You say here that the findings in this paper show a lower proportion of mistreatment than other studies in Ethiopia so this should be mentioned in the abstract rather than what you’ve mentioned about it being higher

7. PLOS authors have the option to publish the peer review history of their article (what does this mean?). If published, this will include your full peer review and any attached files.

Reviewer #3: No

---

## [Author Response · Author response to Decision Letter 2]

7 Mar 2022

To: Editor-in-chief of PLOS ONE Journal

Subject: Submission of a revised manuscript for publication

Dear Editor,

Thank you for allowing us to submit a revised draft of our manuscript entitled “Prevalence and risk factor for mistreatment in childbirth: in health facilities of Gondar city, Ethiopia "[Manuscript ID number: PONE-D-21-04196R2]. We appreciate the time and effort dedicated by you and the reviewers to provide your valuable feedback on our manuscript. We are grateful to the reviewer for their insightful comments which improve our manuscript.

We accepted and tried to incorporate all of the comments provided. Thus, the comments are

attached here below with their point-by-point responses given in blue font color. Besides, the

detailed changes made are highlighted in the “revised manuscript with track changes” to easily identify the changes/improvements, and clean copy of the revised manuscript are prepared.

Response to Reviewer #3 comments 

1.) Suggest editing title to “Prevalence and risk factors for mistreatment in childbirth: in health facilities in Gondar City, Health facilities, Ethiopia

Response: Thank you reviewer for comment. To make easy to reader we modified the title as “Prevalence and risk factor for mistreatment in childbirth: health facilities of Gondar city, Ethiopia”

2.) Abstract-Background section:

a. Is compassionate RMC ensuring women’s survival? I’m unsure of this statement although I do understand that more positive experiences of care could lead to better health outcomes for women and newborns (based on improving the coverage of facility deliveries).

b. You say that compassionate and respectful maternity care has received less attention both in practice and research in Ethiopia I think there is quite a bit of literature about RMC in Ethiopia. Perhaps there is less data from Gondar City specifically and if so, I might state that.

Response for a&b: Thank you reviewer for comments, we have accepted the both comments and we made correction accordingly, please see clean version of manuscript line 15-16.

3.) Abstract—Methods

a. How many facilities was this study conducted in?

Response: Thank you for your clarification question. In this study 9 health facilities are considered. We have stated this in the abstract section, see line 20. 

b. These are postpartum women—how long postpartum? 

Response: Thank you again reviewer for your clarification question. In this study the postpartum period considered was giving birth to 24 hours. 

c. You say: A binary logistic regression analysis was done to see the association between mistreatment and the independent variables. Suggestion: A binary logistic regression analysis was done to see whether there was an association between mistreatment and independent variables such as XXXXXX

d. You say: Finally, the logistic regression analysis was done by stratifying type of party and mode of delivery. I think you mean parity?

Response for c&d: Thank you reviewer for suggestion. We have accepted all the comments and rewrite the paragraph. 

4.) Abstract—Results

a. Suggest changing the word “Result” to “Results” on line 27

b. Line 31—change “complication” to “complications” on line 31

Response a&b: Thank for your comments, we have corrected the comments. Please see line 24 and 29.

c. Did you adjust for facility-based clustering?

Response: Thank your reviewer for clarification question, we did not adjusted for facility based clustering because in this study, all childbirth care providing health facilities; eight health centers, and one referral hospital were considered and we didn’t not use cluster sampling method 

5.) Abstract—Conclusions

a. Suggest capitalizing the word “the” in line 34

b. The conclusion that the proportion of women who were mistreated was higher here than in other developing countries—it seems like it would be useful to compare what you found with other studies in Ethiopia since there have been quite a few published

Response for a&b: Thank you reviewer for your valuable comments, we have made modification, see line 30-31

c. You mention some possible ways to address mistreatment based on your findings. How would you address the issue that mistreatment was more prevalent in hospitals?

Response: Thank you dear reviewer for your question. To address these issues enhancing health workers capacity through trainings like compassionate and respectful maternity care might solves the problem. Please see conclusion section, we have accepted and add the comment in conclusion section. 

d. In your conclusion you state that the proportion of mistreatment was higher in this study but the in the introduction of the paper, you do show ranges of up to 98.9%

Response: Thank you reviewer for your valuable comments, we have made modification, see line 30-31

Introduction

1.Line 43: Maternal mistreatment at childbirth is more than just what you’ve mentioned here. I suggest using a more accepted definition of RMC

Response: Thank you reviewer for your valuable comments, we have made modification, see line 39-42

Methods

2. Line 152: Data were collected through face-to-face interviews. Were these interviews conducted in the health facilities? 

Response: Thank you reviewer for your clarification question. The interview was conducted in the health facilities after giving birth before exit.

2. A chart review was also done but which data/indicators were extracted from the charts?

Response: Thank you for your clarification question. Record reviews were done to extract obstetric related variables such as parity, gravidity, number of antenatal care visits, mode of delivery. We have now incorporated these comments, please see line 153. 

Results

1. Line 203—The majority of participants had ANC visit—I assume this is at least one ANC visit?

Response: Thank you reviewer for your valuable comments, we have made correction accordingly, see line 205

2.Line 245—You state that mothers who had less than 4 ANC visits were 3.58 times more likely to mistreat—but should be changed to experience mistreatment rather than mistreat; this is the same for the next few sentences—the women themselves are not mistreating but are rather being mistreated

3.Line 251—"The analysis was stratified by party…”; this should be parity

Response for 2&3: Thank you reviewer for your suggestion, we have made correction accordingly, see line 247-252

Conclusions:

1.Line 277—You say here that the findings in this paper show a lower proportion of mistreatment than other studies in Ethiopia so this should be mentioned in the abstract rather than what you’ve mentioned about it being higher

Response: Thank you reviewer for your valuable comments, we have made modification, see line 30-31

---

## [Decision Letter · Decision Letter 3]

21 Apr 2022

Prevalence and risk factor for mistreatment in childbirth:  in health facilities of Gondar city, Ethiopia

PONE-D-21-04196R3

Dear Dr. Boke,

We’re pleased to inform you that your manuscript has been judged scientifically suitable for publication and will be formally accepted for publication once it meets all outstanding technical requirements.

Kind regards,

Orvalho Augusto, MD, MPH

Academic Editor

PLOS ONE

Additional Editor Comments (optional):

Reviewers' comments:

Reviewer's Responses to Questions

**Comments to the Author**

1. If the authors have adequately addressed your comments raised in a previous round of review and you feel that this manuscript is now acceptable for publication, you may indicate that here to bypass the “Comments to the Author” section, enter your conflict of interest statement in the “Confidential to Editor” section, and submit your "Accept" recommendation.

Reviewer #3: (No Response)

2. Is the manuscript technically sound, and do the data support the conclusions?

Reviewer #3: Yes

3. Has the statistical analysis been performed appropriately and rigorously? 

Reviewer #3: I Don't Know

4. Have the authors made all data underlying the findings in their manuscript fully available?

Reviewer #3: (No Response)

5. Is the manuscript presented in an intelligible fashion and written in standard English?

Reviewer #3: Yes

6. Review Comments to the Author

Reviewer #3: Thank you for making the revisions on the paper. I just have two additional comments in response to your revisions:

Comment 1: Suggest a minor edit to the current title (“Prevalence and risk factor for mistreatment in childbirth: health facilities of Gondar city, Ethiopia”) to “Prevalence and risk factors for mistreatment in childbirth in health facilities in Gondar City, Ethiopia

Abstract 4c: About sampling the facilities—it’s unclear how you selected these facilities then. Was it purposive? Or did you randomly select from a list?

7. PLOS authors have the option to publish the peer review history of their article (what does this mean?). If published, this will include your full peer review and any attached files.

Reviewer #3: No

---

## [Editor Report · Acceptance letter]

27 Apr 2022

PONE-D-21-04196R3 

Prevalence and risk factor for mistreatment in childbirth:  in health facilities of Gondar city, Ethiopia 

Dear Dr. Boke:

I'm pleased to inform you that your manuscript has been deemed suitable for publication in PLOS ONE. Congratulations! Your manuscript is now with our production department. 

Kind regards, 

on behalf of

Dr. Orvalho Augusto 

Academic Editor

PLOS ONE